

# Tertiary lymphoid structures-driven immune infiltration patterns and their association with survival in neuroblastoma

Xuelian Liu[1,*], Jian Deng[1,2,*], Bingqing Yu[1], Jiaxiong Tan[3], Xiaoliang Lu[1,2] and Minmin Zhang[1,2,4]

[1] Department of Gastroenterology, The First Affiliated Hospital of Jinan University, Guangzhou, China
[2] Department of Hematology, The First Affiliated Hospital of Jinan University, Guangzhou, China
[3] Department of Pediatric Oncology, National Clinical Research Center for Cancer, Key Laboratory of Cancer Prevention and Therapy, Tianjin's Clinical Research Center for Cancer, Tianjin Medical University Cancer Institute and Hospital, Tianjin, China
[4] Department of Emergency, The First Affiliated Hospital of Jinan University, Guangzhou, China
* These authors contributed equally to this work.

## ABSTRACT

**Background:** Neuroblastoma (NB), a diverse childhood cancer, needs better prognostic markers for personalized treatment. The current clinical risk stratification system does not fully explain the high heterogeneity of tumor patients. Tertiary lymphoid structures (TLS), key in tumor immunity, may serve as new biomarkers, but their impact on NB prognosis is unclear.
**Methods:** We combined transcriptome data from NB cohorts GSE49710 and GSE62564, analyzing 37 TLS-related genes. A prognostic signature (CMLS) was created using machine learning and validated with Kaplan-Meier and receiver operating characteristic (ROC) curves. We also studied immune infiltration and gene expression patterns in NB tissues using single-cell sequencing and quantitative real-time polymerase chain reaction (qRT-PCR).
**Results:** A 6-gene TLS signature predicted better survival in NB patients. High levels of CCL2, CCL4, CCL21, CD200, CXCR3, and IGSF6 correlated with improved survival. The low-TLS risk group showed better event-free and overall survival. Immune analysis indicated a higher immune cell presence, especially cytotoxic T cells, in this group. Single-cell sequencing revealed lower TLS gene expression in refractory recurrence samples. CD200 downregulation reduced NB cell invasiveness and migration.
**Conclusion:** Our study demonstrates that TLS-related genes play a crucial role in NB prognosis, with a 6-gene TLS signature (CCL2, CCL4, CCL21, CD200, CXCR3, and IGSF6) serving as a promising prognostic biomarker for NB. CD200 may be a potential target for inhibiting the biological behavior of NB cells.

Corresponding authors
Xiaoliang Lu, lsl123231@163.com
Minmin Zhang,
15692425033@163.com

## INTRODUCTION

Neuroblastoma (NB) is the most common extracranial solid tumor in children, characterized by a wide range of clinical outcomes, from spontaneous regression to aggressive disease progression (*Kennedy et al., 2023*; *Bagatell et al., 2024*). Although current staging systems have improved, they still fail to fully capture the heterogeneity of NB, just as a small number of patients at low to moderate risk still show clinical phenomena of disease progression, which are hard to explain, highlighting the need for novel biomarkers to improve risk stratification and prognostic assessment (*Bagatell et al., 2024*; *Irwin & Goldsmith, 2024*). Immune checkpoint inhibitors targeting PD-1/PD-L1 and CTLA-4 have demonstrated limited therapeutic efficacy in NB (*Mao, Poimenidou & Craig, 2024*). At present, only immunotherapy drugs targeting the GD-2 monoclonal antibody have been approved for the treatment of refractory recurrent NB. In the largest reported series ($n = 1,183$) of high-risk neuroblastoma (HR-NB) patients treated with dinutuximab, isotretinoin, IL-2, and GM-CSF (sargramostim) post-consolidation/maintenance, the 5-year event-free survival rate only reached 61% (*Mora et al., 2024*; *Desai et al., 2022*).

Recent advances in cancer immunotherapy have underscored the importance of the tumor microenvironment (TME), particularly the role of tertiary lymphoid structures (TLS), in modulating antitumor immunity (*Teillaud et al., 2024*). TLS are ectopic lymphoid structures that develop in chronic inflammatory and cancerous tissues and have been linked to improved survival in multiple malignancies (*Teillaud et al., 2024*; *Dong, Wang & Wu, 2023*). Some studies suggest that TLS is closer to the lesion than secondary lymphatic organs, which is conducive to a faster immune response (*Schumacher & Thommen, 2022*). TLSs bear notable anatomical and functional resemblances to secondary lymphoid organs (SLOs), such as lymph nodes and Peyer's patches, yet they lack an encompassing capsule (*Schumacher & Thommen, 2022*). The absence of this capsule may permit their cellular constituents to directly infiltrate the surrounding tissues, which could be essential for recruiting lymphocytes and maintaining an optimal immunological microenvironment (*Ruddle, 2014*; *Sautes-Fridman et al., 2019*). A recent study employed a 9-gene TLS signature to evaluate TLSs in breast cancer (BC), revealing that TLS presence correlated with early TNM stage and improved prognosis in BC patients (*Wang et al., 2022*). Recent research by *Rothe et al. (2025)* revealed that TLS were present in all ganglioneuroma (GN) cases, most ganglioneuroblastoma (GNBL) cases, and a minority of neuroblastoma (NBL) specimens. Notably, their study demonstrated that the presence of TLS in primary neuroblastic tumor (pNT) patients was significantly associated with prolonged progression-free survival, in contrast to all other analyzed immunological markers (*Rothe et al., 2025*).

TME is a sophisticated network comprising immune cells, blood vessels, extracellular matrix, and diverse stromal cells, all of which play a pivotal role in tumorigenesis, therapeutic resistance, and disease progression (*Peng et al., 2024*). TME dysfunctions (*e.g.*, hypoxia, acidic pH, elevated interstitial pressure) can reduce the efficacy of cancer therapies, especially immunotherapies (*Peng et al., 2024*; *Tiwari, Trivedi & Lin, 2022*). Research indicates that normalizing the TME can enhance the efficacy of various cancer

treatments, including chemotherapy, radiotherapy, targeted therapies, and immunotherapies (*Tiwari, Trivedi & Lin, 2022*; *Huang et al., 2022a*, *2022b*). In the context of most solid tumors, the presence of TLS correlates with a decreased likelihood of recurrence and heightened responsiveness to immune checkpoint inhibitors (ICB) (*Meylan et al., 2022*). Studies have also shown significant differences in the frequency with which TLS is detected in the tissues surrounding the tumor and within the tumor. High peritumoral-TLS density and low tumor stroma percentage (L-TSP) are considered independent and favorable prognostic factors for non-metastatic colorectal cancer (nmCRC) patients (*Wang et al., 2022*). Anlotinib has been identified as facilitating the normalization of tumor vasculature, potentially through the activation of CD4+ T cells, which remodels the suppressive TME into a stimulatory one, thereby significantly curbing tumor growth and preventing systemic immunosuppression (*Xu et al., 2022*; *Lou et al., 2024*). Consistent with these findings, reinventing TME and improving TLS ratio may enhance the efficacy of immunotherapy in neuroblastoma and improve patient outcomes.

In this study, we integrated multicenter NB datasets to develop and validate a TLS-related gene signature through a consensus machine learning framework. We hypothesized that this signature would serve as a prognostic biomarker and reveal interactions between TLSs, immune infiltration, and survival. Furthermore, we compared TLS-related gene expression in immune cells from newly diagnosed *vs.* relapsed NB patients using single-cell RNA sequencing of 17 NB tissues. Finally, we demonstrated that CD200 downregulation inhibits NB cell invasion and migration.

## MATERIALS AND METHODS

### Data preprocessing of multicenter cohorts

Comprehensive neuroblastoma data were meticulously assembled from two multicenter cohorts, leveraging datasets available in the Gene Expression Omnibus (GEO, accessible at http://www.ncbi.nlm.nih.gov/geo), with specific reference to GSE49710 and GSE62564. Additionally, single-cell sequencing data from neuroblastoma tissues were included from the GSE147766 dataset. These rich datasets provide a robust foundation for our analysis.

### Univariate Cox regression

Univariate Cox regression analysis performed using R software (version 4.2.1) with the "survival" package was applied to two separate neuroblastoma datasets (GSE49710 and GSE62564) in our study. Forest plots were generated using the "ggplot2" package (version 3.3.6) for result visualization.

### Establishment of a consensus machine learning-driven prognostic signature

To evaluate the consensus machine learning-driven signature (CMLS) and its association with patient prognosis, GSE49710, which contains detailed clinical information, was used as the training set, while GSE62564 served as the validation set. This approach enabled robust assessment of the CMLS's predictive accuracy across different patient groups. We integrated insights from ten machine learning algorithms, including CoxBoost, stepwise
Cox regression, Least Absolute Shrinkage and Selection Operator (LASSO), Ridge regression, Elastic Net, survival support vector machines, generalized boosted regression models, and supervised principal component analysis. This multidisciplinary strategy aimed to enhance the predictive validity and adaptability of the CMLS.

Model optimization for CoxBoost began with the "optimCoxBoostPenalty" function to determine the optimal penalty value, followed by 10-fold cross-validation to identify the ideal number of boosting steps. The "CoxBoost" function was then used for model fitting. Stepwise Cox regression analyses were conducted using the "survival" package, with model complexity assessed *via* the Akaike Information Criterion (AIC). LASSO, Ridge, and Elastic Net models were implemented using the "glmnet" package, with the regularization parameter lambda determined through 10-fold cross-validation. The alpha parameter, ranging from 0 to 1, adjusted the model between LASSO (alpha = 1) and Ridge (alpha = 0) regularization. The Survival-SVM model was instantiated *via* the "survivalsvm" function, while the GBM model fitting utilized the "gbm" function, also optimized through 10-fold cross-validation. The SuperPC model was executed using the superpc package, with cross-validation through the "superpc.cv" function. For the plsRcox model, we employed the "cv.plsRcox" function, and the Random Survival Forest model was delineated using the "rfsrc" function, setting "ntree" to 1,000 and "nodesize" to 5.

## Prognostic value of CMLS and potential clinical application

Scores were assigned to each sample in both training and validation cohorts based on model outputs, categorizing them into high- and low-CMLS groups. The prognostic relevance of the CMLS was assessed using Kaplan-Meier survival curves, and its prognostic accuracy was evaluated through time-dependent receiver operating characteristic (ROC) curves.

## Immune infiltration analysis

In this study, we conducted a comprehensive immune infiltration analysis on 498 neuroblastoma RNA sequencing samples derived from the GSE49710 dataset. To achieve this, we utilized several R packages that are specifically designed for the estimation and visualization of tumor-infiltrating immune cells in gene expression data.

### TIMER

Tumor immune estimation resource (TIMER) is a comprehensive resource for analyzing the abundance of immune cells in diverse cancer types. We employed the 'TIMER' R package to estimate the relative abundance of various immune cell types within the tumor microenvironment of neuroblastoma samples. By leveraging the TIMER package, we were able to assess the immune landscape of the neuroblastoma samples from GSE49710.

### ESTIMATE

To further validate the immune cell infiltration patterns observed with TIMER, we utilized the 'ESTIMATE' R package. Estimation of STromal and immune cells in malignant tumor tissues using expression data (ESTIMATE) is a widely recognized tool for estimating the content of stromal and immune cells in tumor tissues based on gene expression profiles.

We applied this package to our dataset to corroborate the TIMER findings and to gain additional insights into the tumor microenvironment.

### CIBERSORT

For a more granular analysis of immune cell subsets, we employed the 'CIBERSORT' R package. Cell-type identification by estimating relative subsets of RNA transcripts (CIBERSORT) is a deconvolution method that estimates the relative proportions of various immune cell types within complex tissue samples, using gene expression data. We applied CIBERSORT to the GSE49710 dataset to resolve the composition of immune cell subsets.

## EMT, hypoxia status, tumor cell stemness, and angiogenesis score comprehensive analysis

In this study, we performed a thorough examination of epithelial-mesenchymal transition (EMT), hypoxia, tumor stemness, and angiogenesis in tumor samples. We assessed EMT using 200 genes from the HALLMARK_EPITHELIAL_MESENCHYMAL_TRANSITION pathway and calculated EMT scores for each sample *via* single-sample Gene Set Enrichment Analysis (ssGSEA). Hypoxia was evaluated with ssGSEA on genes from the HALLMARK_HYPOXIA pathway to assign hypoxia scores. We broadened our analysis to include KEGG-listed pathways to understand the activity of various signaling pathways within tumors, calculating their activity scores with ssGSEA. For tumor cell stemness quantification, we utilized human stem cell datasets from the Progenitor Cell Biology Consortium (PCBC) on Synapse.org and applied the One Class Linear Regression (OCLR) method to measure stemness levels.

## Real-time quantitative PCR and transwell experiment

NB cell line was purchased from Meisen Cell Biotechnology Co., LTD., Zhejiang, China. For the primer sequences corresponding to each gene in qPCR experiments, detailed information can be found in Table S1 (submitted). Both the SK-N-BE (2) and SH-SY5Y neuroblastoma cell lines were amplified using standard cell culture techniques. In order to compare the relative expression levels of the two cell lines, GAPDH was used as the internal reference gene, and the other six genes were used as the experimental group. First, the culture solution of the petri dish was sucked out, and then the cells were cleaned once with $1 \times$ PBS and the adherent NB cells were removed with a cell scraper. After adding 1 mL of RNAex and shaking well, all liquids were transferred to an enzyme-free centrifuge tube for centrifugation, and placed in an ice box for 5 min of precipitation to prepare for subsequent purification. Before extraction and purification, 1 ml cold ethanol of equal volume (70% concentration) was added, mixed, transferred to RNAmini column and centrifuged at 12,000 rpm for 1 min. Discard the bottom liquid that has been centrifuged, add 600 ul Buff RW1 and mix well. Then centrifuge under the same conditions for 1 min and remove the bottom liquid. Next add 650 ul Buff RW2 to clean twice, then transfer to the collection tube with column core and centrifuge. After removing the lower liquid, the column core was transferred to a new collection tube, and 40 ul of enzyme-free water was

added to the middle and placed on ice for 30 min. Centrifuge at 12,000 rpm for 3 min, and then the RNA concentration was measured. According to the indication of the reverse transcription kit we used, the RNA concentration should not be lower than 70 ng/mL. mRNA was extracted using a commercial kit (Total RNA Purification Kit) and quantified with a spectrophotometer. This kit was purchased from Accurate Biotechnology (Hunan, China) Co., Ltd. Using a NanoDrop spectrophotometer, a blank measure is first used and then the concentration of the sample is tested. The ratio of A260/A280 was calculated, and the ratio was ≥1.8, which met the experimental requirements. According to the instructions, 30 ul system is used for reverse transcription. According to the instructions of the RNA-to-cDNA™ Kit(Applied Biosystems™), firstly, add 6 ul of 5× primescript buffer and 1.5 ul of ENZYME MIX, Olido dT primer and Random 6 mers each. Then add no more than 19.5 ul of RNA samples. Finally, RNA-free H2O was supplemented to make the system reach 30 ul. Reaction conditions: Reverse transcription temperature 42 degrees Celsius for 15 min, inactivation temperature 85 degrees Celsius for 5 min total of 20 min. The 20 ul qPCR system was respectively added with: 10 ul of 2X SYBR Green Pro Taq HS Premix, 2 ul of CDNA, 1 ul each of the forward primer and reverse primer, and 6 ul of RNAse free H2O, according to the reagent kit (Accurate Biology, Guangzhou, China). RT-PCR experiments were conducted on the Bio-Rad CFX Manager platform. The GAPDH gene served as an internal reference gene, with each target gene having three separate reaction wells to ensure the accuracy of the results.

si-CD200 was designed and customized according to the siRNA sequence. siRNA was diluted to an appropriate concentration (20 nM) and mixed with Lipofectamine RNAiMAX transfection reagent, and the transfection complex was added to the culture dish of NB cells and incubated for 48 h. The concentration of si-CD200 is based on the possible optimal concentration range obtained from the previous concentration gradient pre-experiment, which is between 10 and 30 nM. The culture medium was DMEM medium with 10% fetal bovine serum (FBS) and 1% penicillin/streptomycin (P/S) mixture. Incubation was carried out in an incubator at 37 °C and 5% CO2. An appropriate amount of basement membrane (Matrigel) is applied to the porous membrane of the Transwell chamber and hydrated to simulate the extracellular matrix. Place the Transwell chamber into a 24-well plate. The cell suspension including $1 \times 10^5$ was added to the upper chamber of Transwell chamber and the medium was added to the lower chamber. The Transwell chamber is cultured in a cell incubator to give the cells time to migrate or invade through the porous membrane. After culture, migrating or invading cells were immobilized with paraformaldehyde and then stained with crystal violet stain. By looking under a microscope and counting the cells that migrate to the underside of the membrane.

## Single cell sequencing analysis and correlation analysis of drug sensitivity

To begin with, we implemented stringent quality assurance measures on the acquired GSE147766 dataset. This was essential to guarantee the integrity and dependability of the data. In this process, we evaluated the sequencing depth to ensure it met coverage requirements, discarded cells that did not comply with quality criteria, and eliminated cells

containing a disproportionately high percentage of mitochondrial genes (Fig. S1). To perform principal component analysis (PCA) on our preprocessed dataset, we utilized the RunPCA function from the Seurat package. We then reduced the dimensionality by selecting high variable genes (HVGs) and focusing on the eighth principal component (PC8). Following this, we employed the Rtsne function to conduct t-distributed stochastic neighbor embedding (t-SNE) clustering analysis. Finally, we annotated the clustered cell populations using CellMarker 2.0 to identify the distinct cellular components. After annotation, the expression of target genes in UMAP is visualized and grouped using violin maps. Finally, we divided 17 NB samples into newly diagnosed group (14 NBs) and refractory relapse group (3 NBs) according to clinical data for single cell level visual analysis.

Furthermore, the relationship between the mRNA levels of the six candidate genes and the sensitivity to common anticancer drugs was analyzed using pan-cancer data from the Genomics of Drug Sensitivity in Cancer (GDSC) and Cancer Therapeutics Response Portal (CTRP). These analyses were performed on the Gene Set Cancer Analysis (GSCA) platform at wchscu.cn.

## Statistical methods

We conducted statistical analyses and generated graphs utilizing R software (version 4.3.3). To evaluate the differences between the two groups, we employed both two-tailed unpaired Student's t-tests and Wilcoxon rank-sum tests. For the comparison of categorical variables, Fisher's exact test was employed. The threshold for statistical significance was established at $P < 0.05$. The above statistics were analyzed using SPSS 22.0 software (IBM Corp., Armonk, NY, USA).

# RESULTS

## Construction of an efficient prognostic model based on tertiary lymphoid structure-related genes in neuroblastoma

We conducted a thorough literature review to identify genes associated with TLS in cancer, yielding 37 TLS-related genes (Sautes-Fridman et al., 2019). This gene set encompasses 12 chemokine signature genes, eight T follicular helper (TFH) cell signature genes, 15 T helper 1 (TH1) cell and B cell signature genes, and two plasma cell and CXCL13 signature genes (Sautes-Fridman et al., 2019). As depicted in Figs. 1A, 1B, we performed univariate COX regression analysis on the expression levels and prognostic data of these 37 TLS-related genes in two neuroblastoma transcriptome datasets, GSE49710 and GSE62564. Genes to the left of the forest plot line with a p-value less than 0.05 were considered to be NB prognostic TLS-related genes.

## Establishment of a consensus machine learning-driven prognostic signature

We utilized the GSE49710 dataset as the training set and the GSE62564 dataset as the validation set to incorporate the prognostic TLS-related genes identified through univariate COX regression analysis into a consensus machine learning-driven algorithm comprising ten distinct machine learning methods. The goal was to select the machine

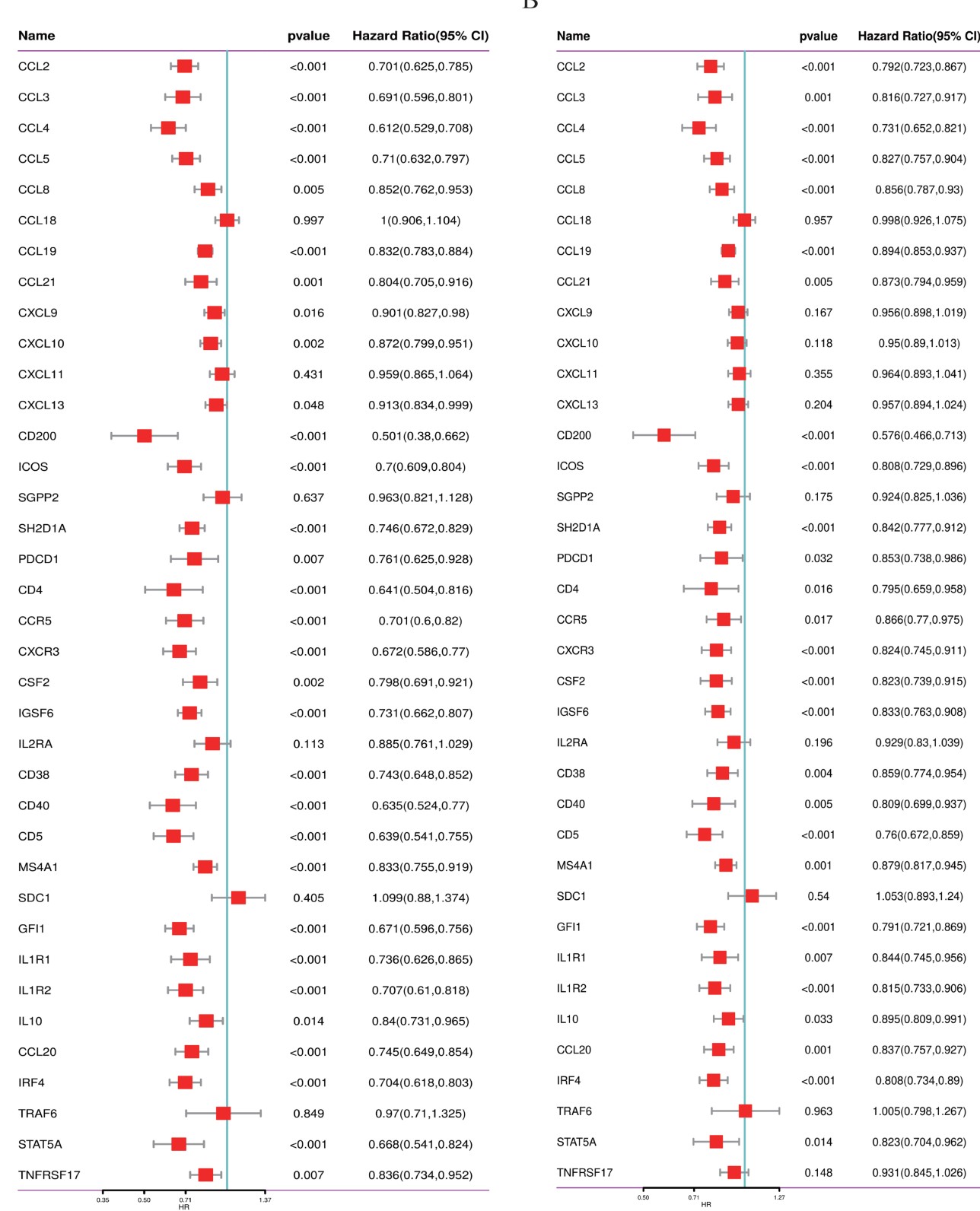

**Figure 1 Screening of prognostic genes related to the tertiary lymphoid structure of neuroblastoma.** (A) Forest plot of univariate COX regression analysis based on the GSE49710 dataset. (B) Forest plot of univariate COX regression analysis based on the GSE62564 dataset.

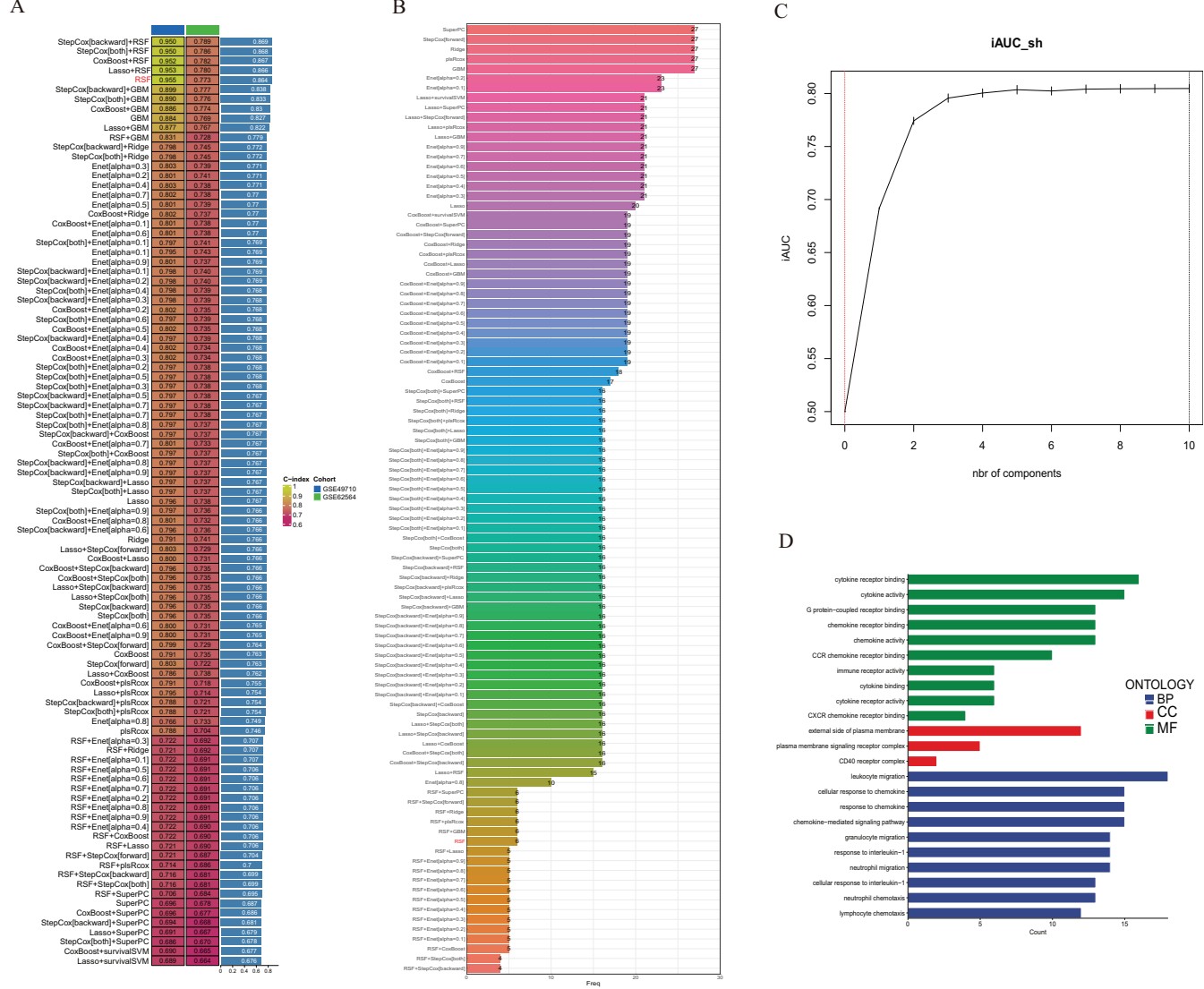

**Figure 2 Construction of consensus machine learning-based prognostic signature.** (A) A combination algorithm of 10 machine learning methods, the blue column represents the results for GSE49710, the green column corresponds to the results for GSE62564, and the third column corresponds to the C-index. (B) It displays the number of features each algorithm combination suggests to include. (C) For the constructed characteristic curve, the horizontal coordinate is the characteristic number and the vertical coordinate is the AUC value. (D) BP, CC and MF represent biological processes, cellular components and molecular functions, respectively.

learning method with the highest concordance index (C-index) to construct the prognostic model. As shown in Fig. 2A, the random forest model ranked fifth with the highest C-index of 0.955. Analysis of the optimal number of features to include, as recommended by different machine learning combinations, revealed that the random forest model achieved the best accuracy when six feature genes were included (Fig. 2B). The area under the curve (AUC) also indicated that the performance plateaued when six features were incorporated (Fig. 2C). Consequently, we developed a prognostic scoring model based on six TLS-related genes: Risk score = (−0.274148539440277 * expression of CCL4) +

(−0.503939241290335 * expression of CD200) + (−0.356039730964334 * expression of CXCR3) + (−0.109685444318236 * expression of CCL2) + (0.195937517824706 * expression of CCL21) + (0.0498379732890246 * expression of IGSF6).

Gene Ontology (GO) analysis revealed that these differentially expressed genes are enriched in multiple pathways involving cytokine and chemokine receptor activation, and they participate in the composition of membrane signal receptor complexes, mediating leukocyte migration and various chemokine responses (Fig. 2D).

## Association of TLS-related genes and the constructed model with neuroblastoma patient survival

To validate the efficacy of the feature genes selected by machine learning, we employed the GES49710 dataset as a training set for single-gene efficacy validation. As shown in Figs. 3A–3F, high expression levels of CCL2, CCL4, CCL21, CD200, CXCR3, and IGSF6 were significantly associated with improved survival in neuroblastoma patients (all $p < 0.001$). These trends were also observed in the independent external validation set GSE62564 ($p < 0.001$, $p < 0.001$, $p = 0.007$, $p < 0.001$, $p = 0.005$, and $p = 0.002$, respectively) (Figs. 3G–3L). Subsequently, we further validated the prognostic power of the TLS gene score model in predicting neuroblastoma event-free survival (EFS) and overall survival (OS). Results in Figs. 4A, 4B, 4G and 4H demonstrated that the low-risk score neuroblastoma group exhibited significantly better EFS and OS in both the training and validation sets (all $p < 0.001$). Diagnostic ROC curves confirmed the predictive efficacy of the score model for OS and EFS (AUCs of 0.943 and 0.747; 0.795 and 0.681, respectively) (Figs. 4C, 4D, 4I, 4J). Time-dependent ROC curves also showed that the score model had high predictive efficacy for OS and EFS at 0.5 and 1 year (AUCs ranging from 0.611 to 0.979) (Figs. 4E, 4F, 4K, 4L).

## Immune infiltration scores

We employed multiple assessment methods, including TIMER, ESTIMATE, and CIBERSORT, to analyze differences in immune cell infiltration between groups with high and low model scores. Figure 5A presents the relative abundance of six immune cell types: B cells, CD4+ T cells, CD8+ T cells, macrophages, neutrophils, and dendritic cells, indicating that the low model score group has a higher abundance of various immune cells. ESTIMATE results revealed that the low-score neuroblastoma group had higher stromal, immune, and ESTIMATE scores, implying a lower tumor cell purity (Figs. 5B–5D). In contrast, the CIBERSORT algorithm score showed that the low-score group had a higher proportion of Tregs and lower proportions of CD8+ T cells and NK cells. Consistently, the distribution of macrophages and B cells was similar to the other two immune scores, with the low-score group having a higher proportion of high-function immune cells such as M1 macrophages (Fig. 5E). The overall correlation analysis is displayed in Fig. 5F.

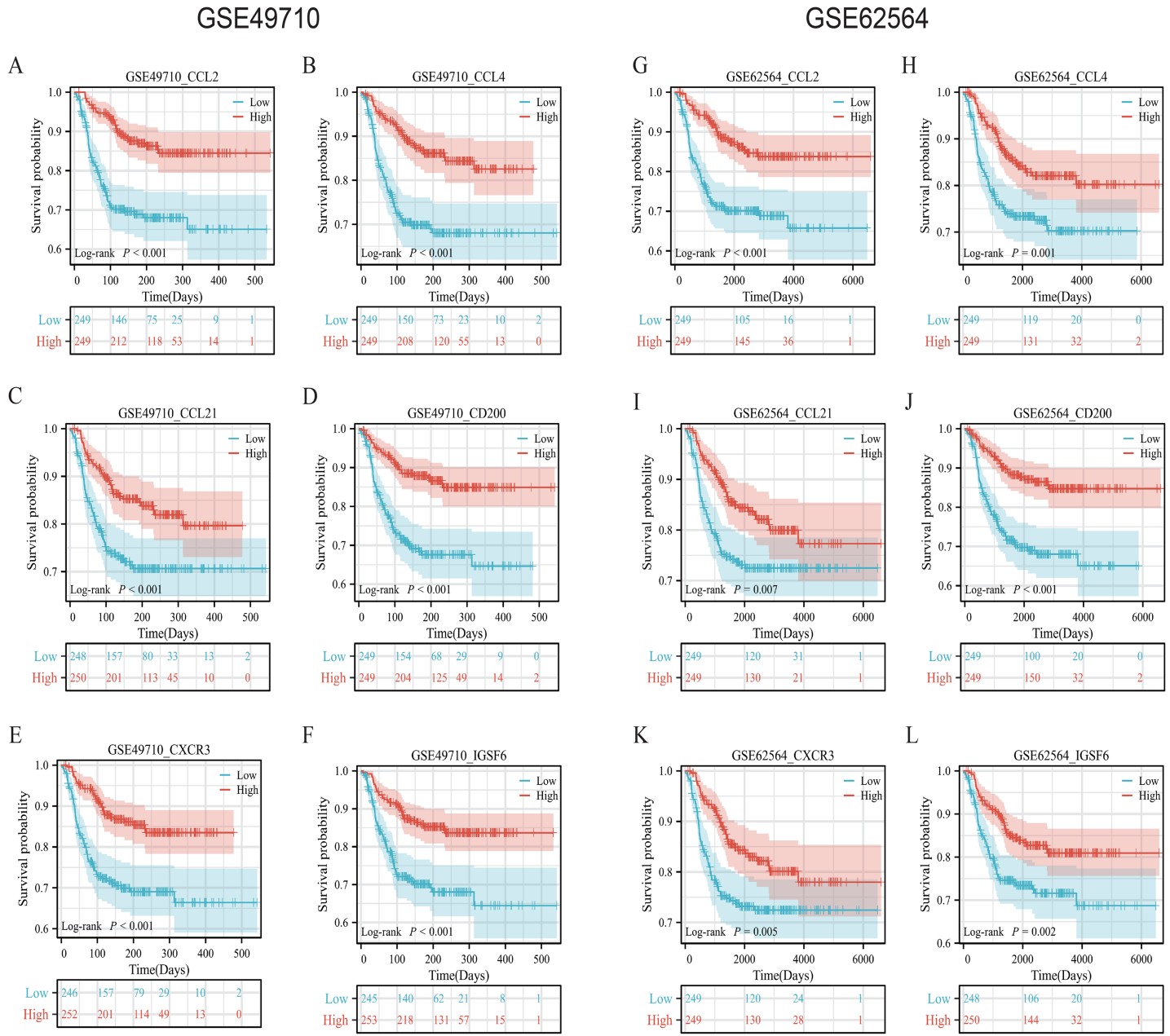

**Figure 3 Survival analysis of candidate genes in NB samples.** (A–F) was K-M analysis of six candidate genes in the GSE49710 dataset, respectively. (G–L) was the K-M analysis of six candidate genes in the GSE49710 dataset, respectively. The red curve represents the group with high gene expression and the blue curve represents the group with low gene expression.

## Comprehensive scores for EMT, hypoxia, angiogenesis, tumor stemness, and signal transduction pathways

EMT, hypoxia, angiogenesis, and tumor stemness are key indicators for evaluating tumor recurrence, metastasis, and patient prognosis. We calculated these scores using the ssGSEA algorithm, and the results, as shown in Fig. 6, indicate that the neuroblastoma group with lower model scores has higher EMT, hypoxia, and angiogenesis scores (Figs. 6A–6C,

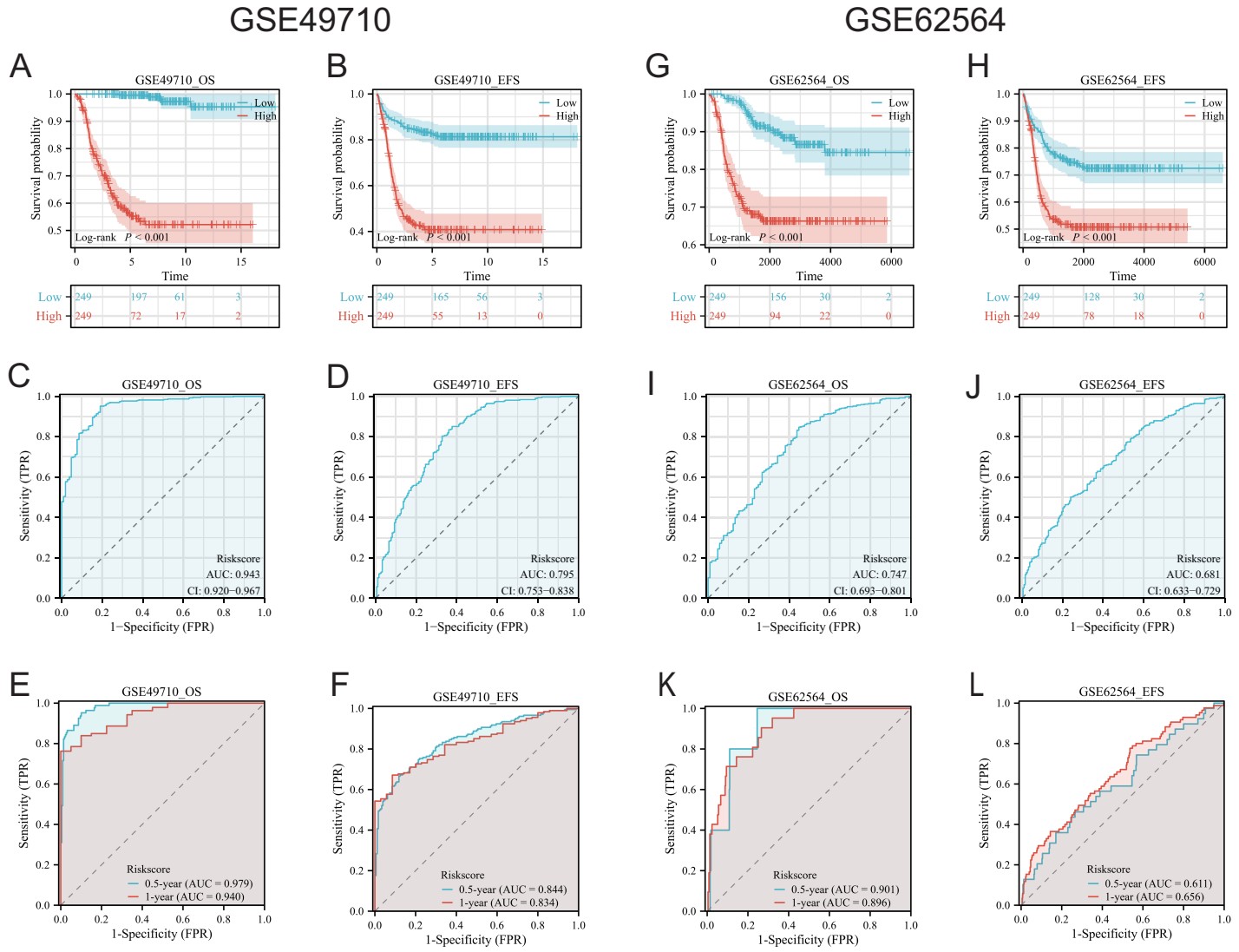

**Figure 4 Comprehensive analysis of the performance of TLS scoring model.** (A and B), (G and H) are K-M analysis curves of OS and EFS of TLS scoring model, respectively. The red curve represents NB groups with high ratings and the blue represents NB groups with low ratings. (C and D) and (I and J) are the diagnostic dependent ROC curves of the model respectively, and the gray shadow area corresponds to the AUC value. (E and F) and (K and L) were the time-dependent ROC curves of 0.5 and 1 year of the scoring model, respectively. Blue corresponds to 0.5 years, red corresponds to 1 year of survival.

6E–6G) and lower tumor stemness scores (Figs. 6D, 6H). Tumor-related signal transduction pathway scores revealed that the group with lower scores has higher scores for various signal transduction pathways (Figs. 7A, 7B), which is consistent with the results of KEGG and GO pathway enrichment (Fig. 7C).

## Clinical feature analysis and nomogram construction

Subsequently, we conducted a detailed analysis of the relationship between the scoring model and clinical characteristics. As shown in Fig. 8A, patients with International Neuroblastoma Staging System (INSS) stages 3 and 4 had higher scores, which is

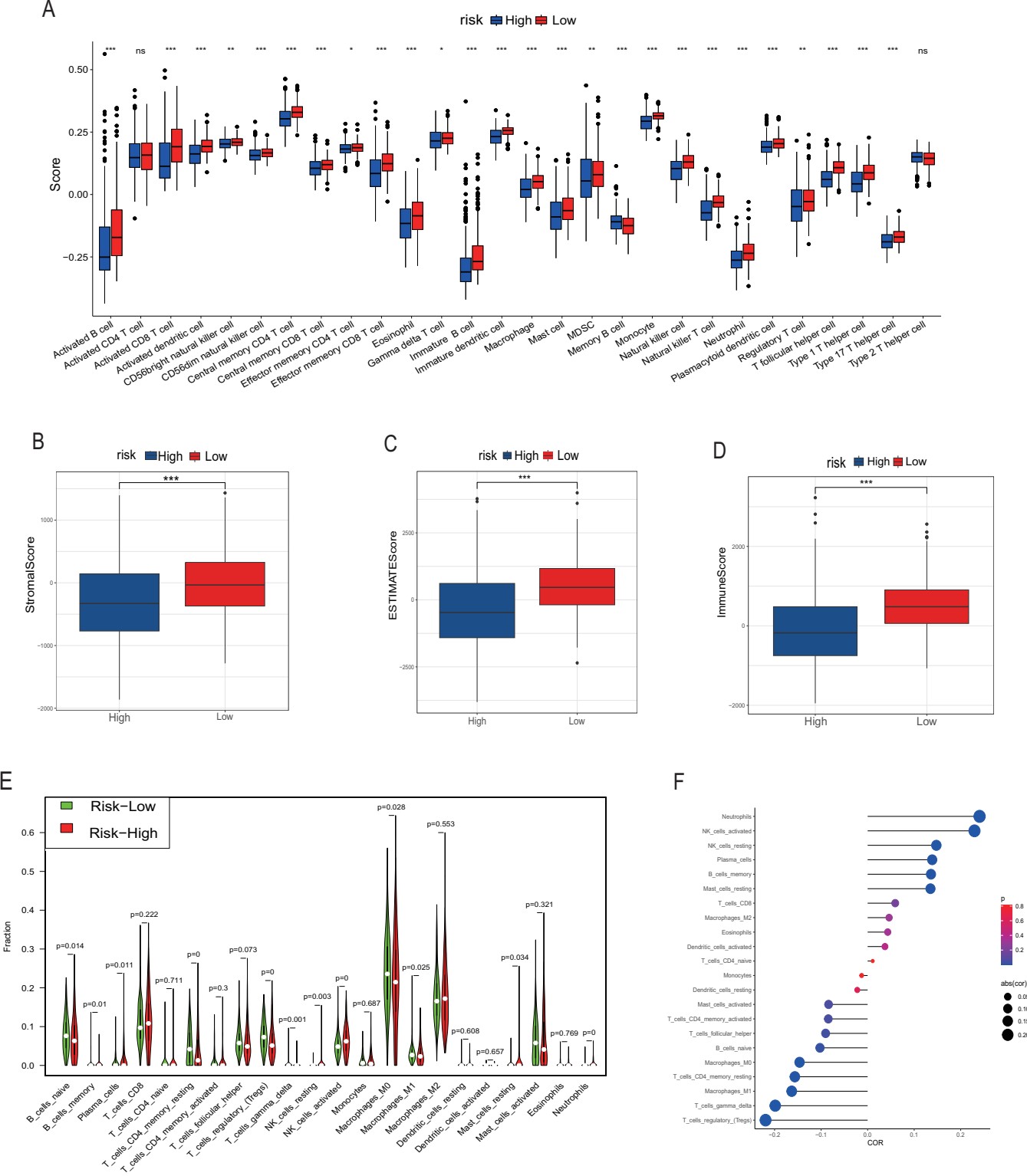

**Figure 5 Analysis of immune infiltration.** (A, B–D) and (E–G) correspond to the analysis results of TIMER, ESTIMATE and CIBERSORT, respectively. The red column represents the low rating group, and the blue represents the high rating group in (A–D). In (E), red represents the high rating group and green represents the low rating group. The size of the circle in the F diagram represents the degree of correlation. *, **, *** represent $P < 0.05$, $P < 0.01$, $P < 0.001$, respectively.

## GSE49710

## GSE62564

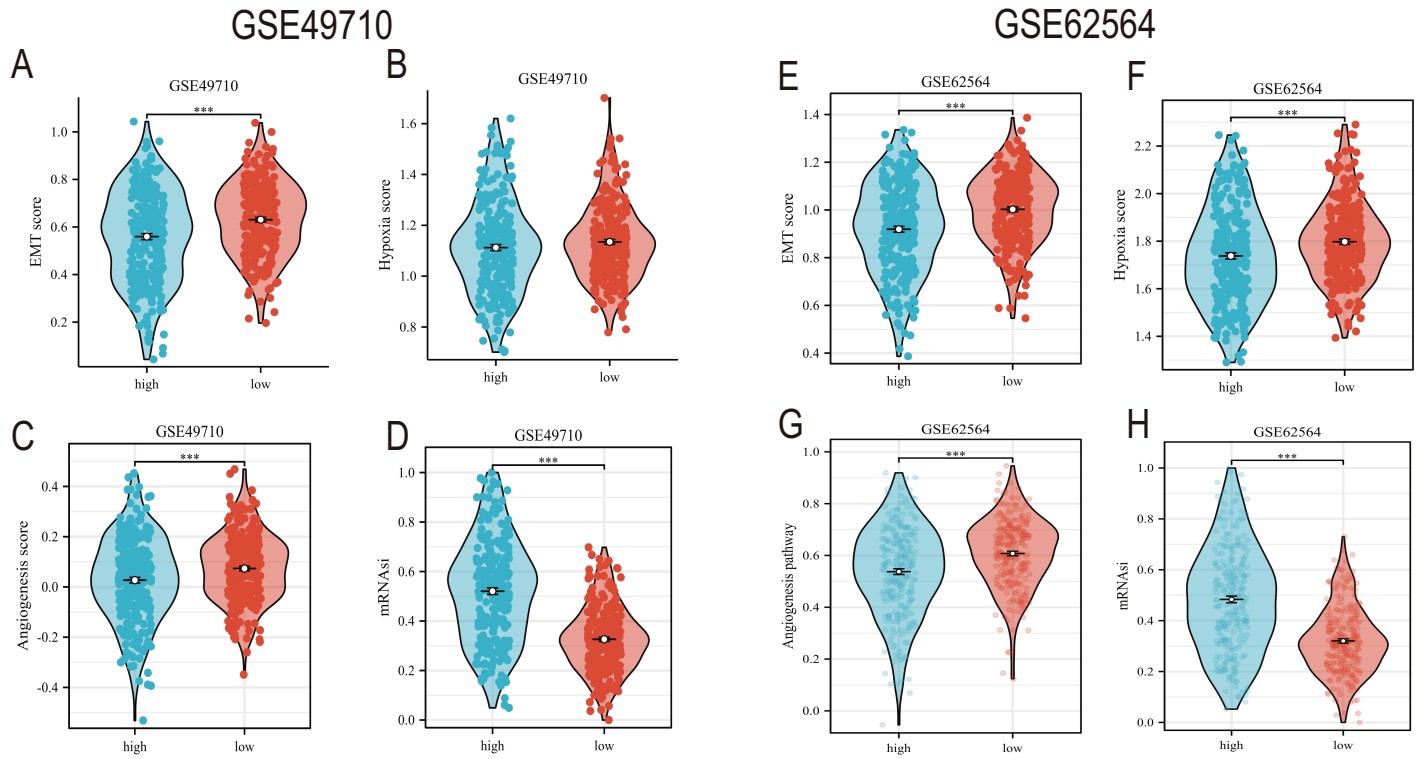

**Figure 6 The correlation analysis between the TLS model and TME, hypoxia scores, vascular scores and tumor stemness.** (A–D) and (E–H) showed the relationship between TLS and EMT scores, hypoxia scores, vascular scores and tumor stemness in GSE49710 and GSE62564 datasets, respectively. *** represent *p* < 0.001.

consistent with the higher scores observed in clinically high-risk and deceased patient groups (Figs. 8B, 8D). MYCN amplification is known to predict poor prognosis in neuroblastoma (NB) patients, and our subgroup analysis revealed that the NB group with MYCN amplification had higher scores (Fig. 8C). Given the potential association between tertiary lymphoid structure formation and immune cell function, we analyzed the relationship between model scores and cytotoxic factors. The results, as depicted in Fig. 8E, showed that the low-score NB group had higher levels of GZMB, PRF1, IFNG, IL-2, IL-15, and IL-18. Based on these findings, we included multiple clinical characteristics such as gender, age, staging, and MYCN amplification in univariate COX regression and constructed nomograms suitable for clinical practice that incorporated factors significantly related to NB prognosis (Figs. 9A, 9B). Subgroup analysis of the newly constructed clinical characteristic scoring model revealed that the low-score group had significantly longer survival times compared to the high-score group (Fig. 9C).

## qRT-PCR and transwell experiment of NB cell lines

In order to detect the difference in expression of six TLS signature genes in non-MYCN amplified and MYCN amplified cell lines, we selected SH-SY5Y and SK-N-BE (2) for real-time quantitative PCR detection after conventional cell culture amplification. Results as shown in Figs. 10A–10F, after three multiple hole repeats, except for the relative
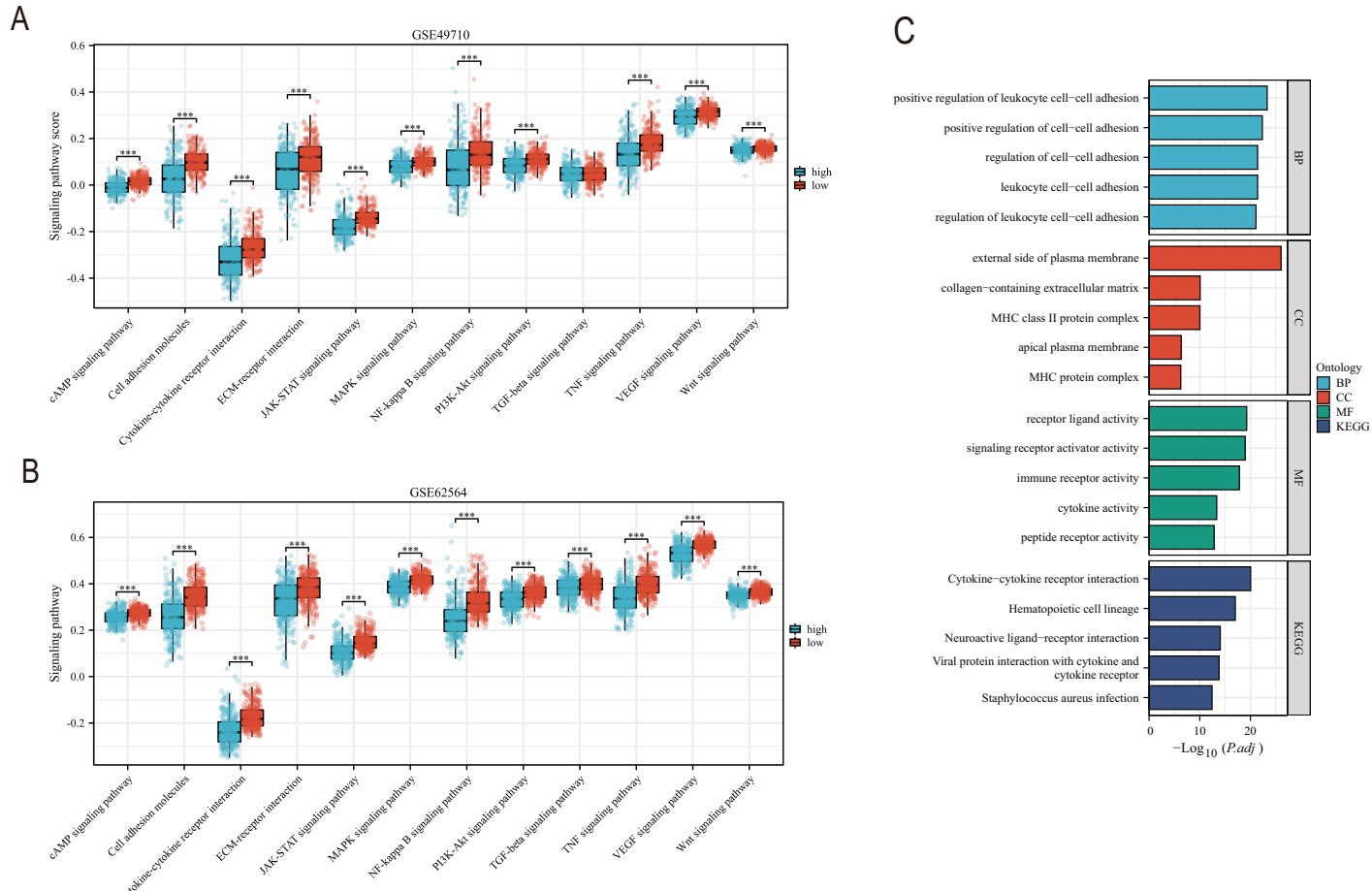

**Figure 7 The correlation analysis between the TLS model and signaling pathways.** (A and B) is the score difference between high and low rating groups in multiple signaling pathways. (C) is enrichment analysis of GO and KEGG. Blue columns correspond to high TLS scores and red columns correspond to low TLS scores. *** represent $p < 0.001$.

expression levels of CCL21 and IGSF6 increased in SH-SY5Y cell line, the other four characteristic genes were significantly lower than SK-N-BE (2) cell line. Figures 10G–10I shows the difference in the expression of each characteristic gene relative to the mean ct value of GAPDH gene in two different cell lines. Except for IGSF6, the expression level of other characteristic genes in SK-N-BE (2) was higher than that of SH-SY5Y. Considering that CD200 was assigned the largest coefficient in our model and that we found the phenomenon of CD200 aggregation and expression in tumor cells through single-cell cluster analysis, we decided to study the effect of CD200 on NB cells. Next, we down-regulated CD200 expression in NB cells using small interfering RNA, and then verified the effect of CD200 on invasion and migration of NB cells. As shown in Figs. 10M–10R, the invasion and migration ability of both NB cell lines decreased significantly than WT group and NC group after interference with CD200.

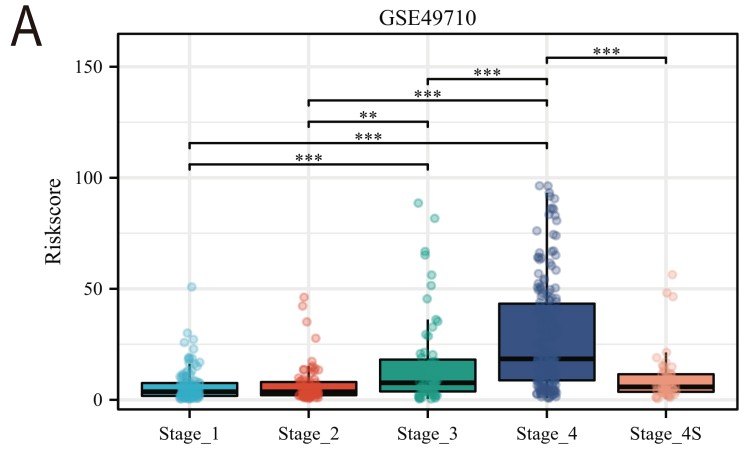

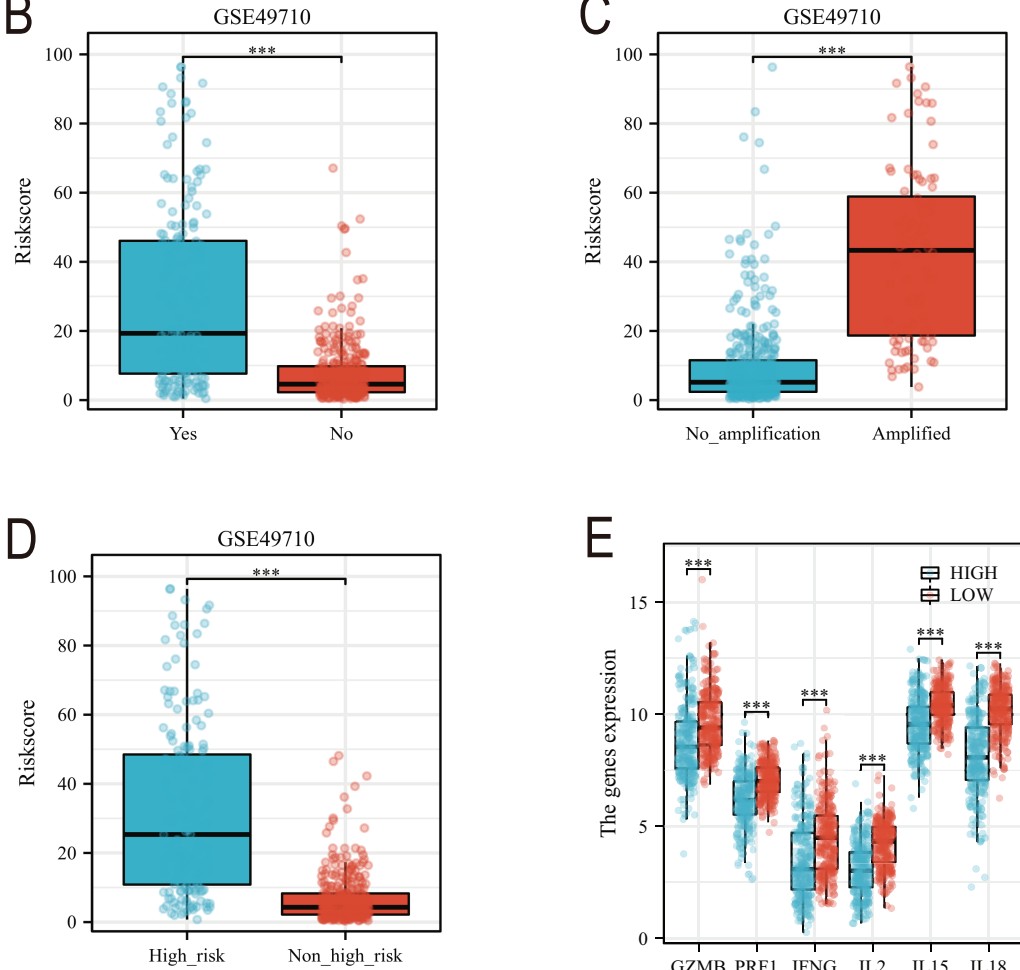

**Figure 8 Analysis of clinical characteristics of TLS model.** (A–E) represent subgroup analyses of the TLS scoring model in different INSS stage NB samples, NB samples with or without death, samples with or without MYCN amplification, clinical risk differences, and cytotoxic cytokines. **, *** represent *P* < 0.01, *P* < 0.001, respectively.

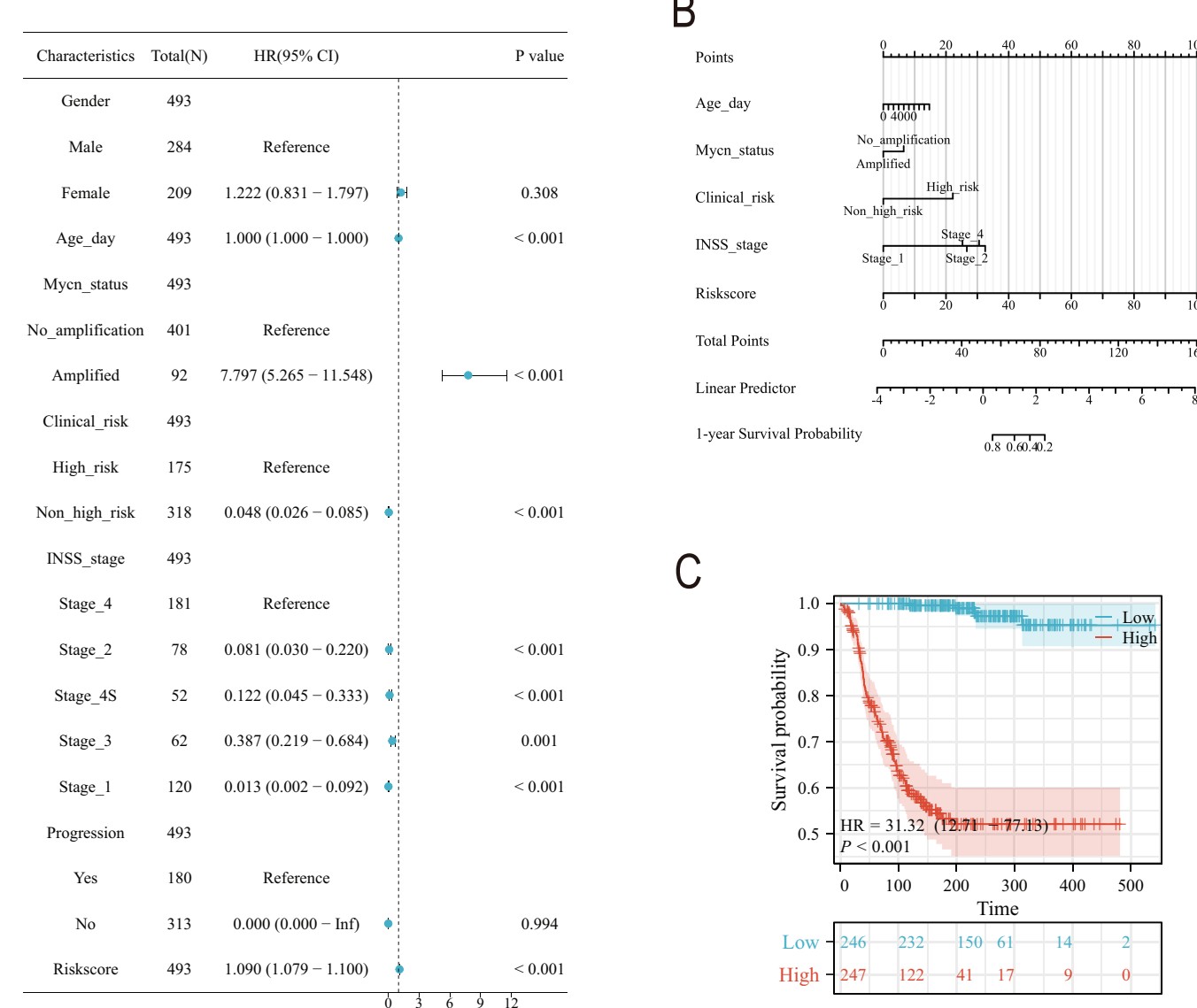

**Figure 9 The construction of Norman map.** (A) is the unifactor COX analysis of clinical features. (B) is the Norman map constructed by integrating clinical factors, and (C) is the K-M curve of the clinical features model.

## Single-cell sequencing analysis of NB samples

To gain a deeper understanding of the expression patterns of TLS model-related genes across different immune cells, we performed tSNE and UMAP clustering analysis and annotation on single-cell sequencing samples from 17 NB tissues (Figs. S3A–S3D). Figure S3E presents a bar chart showing the proportion distribution of various immune cells in the 17 NB samples. Further single-gene UMAP analysis revealed that CCL2 and CCL21 are primarily distributed in smooth muscle cells, tissue stem cells, endothelial cells, and a subset of macrophage subpopulations. CCL4 and CXCR3 are mainly found in NK

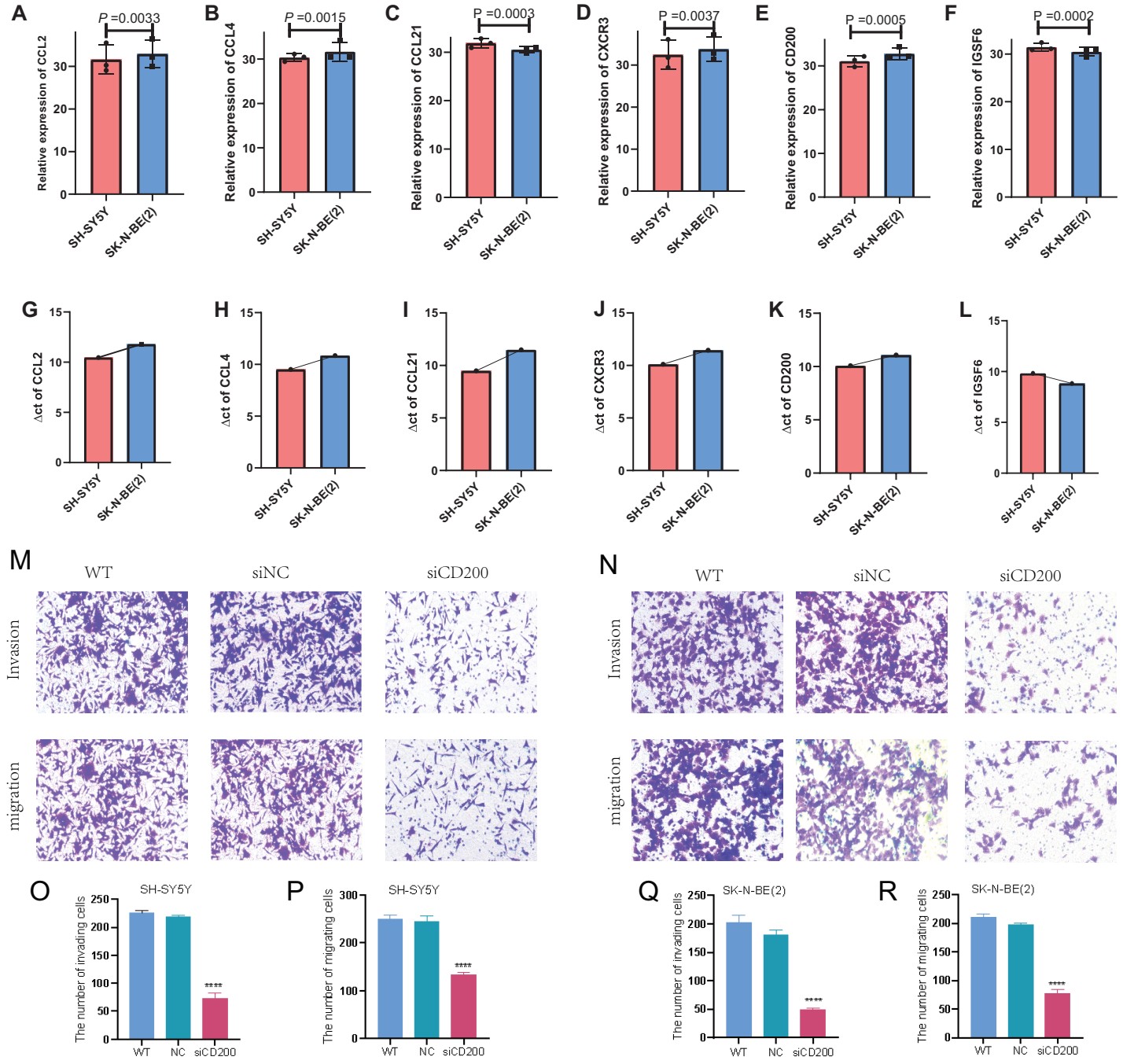

**Figure 10  qRT-PCR and Transwell experiment of NB cell lines.** (A–F) Average expression levels of 6 TLS related genes in two NB cell lines. (G–L) The ratio of the expression level of each gene to the internal reference gene (GAPDH). The red column represents SH-SY5Y and the blue column represents SK-N-(BE) 2. M-R shows the invasion and migration results after WT, NC, and siRNA interference with CD200.

cells, T cells, and monocyte-macrophage subpopulations. CD200 was expressed in various immune cells to varying degrees. In our single-cell cluster analysis, we found a very interesting phenomenon: CD200 was expressed at a higher level in neuron-like tumor cell

subpopulations. IGSF6 is mainly concentrated in monocyte-macrophage subpopulations (Figs. S3F–S3Q). Subsequently, we conducted subgroup comparative analysis on single-cell samples from three refractory relapsed NB cases and 14 newly diagnosed NB cases (Figs. S4, S5). Figures S4, 5A–5D show the cellular distribution maps after tSNE and UMAP clustering analysis and annotation of RR-NB and ND-NB single-cell samples. Figures S4, S5E presents the statistical results of cellular distribution differences in different NB subgroup samples. It is evident that, compared to most ND-NB samples, RR-NB samples have a higher proportion of neuron-like NB cells and a lower proportion of T cells and B cells (Figs. S4, S5E). Single-gene UMAP distribution and statistical analysis found that the enrichment density of the six characteristic genes in the 14 ND-NB sample cell subpopulations was significantly higher than in RR-NB samples (Figs. S4, S5F–S5Q).

### Correlation analysis of anti-tumor drug sensitivity

Finally, we utilized the Genomics of Drug Sensitivity in Cancer (GDSC) and Cancer Therapeutics Response Portal (CTRP) databases to perform correlation analysis between single-gene expression and drug sensitivity. The results showed that the transcriptomic expression level of CCL2 is significantly positively correlated with the sensitivity to the TOP30 anti-tumor drugs, while the other five genes exhibit varying degrees of negative correlation with most drugs. Notably, the expression levels of IGSF6, CXCR3, and CCL4 genes showed the most pronounced negative correlation with drug sensitivity (Figs. S2A, S2B).

## DISCUSSION

In this study, we identified a 6-gene TLS signature using machine learning algorithms and demonstrated its association with improved survival in NB patients. High expression levels of CCL2, CCL4, CCL21, CD200, CXCR3, and IGSF6 were significantly associated with improved survival in both training and validation sets. The low-risk score NB group exhibited significantly better event-free survival (EFS) and overall survival (OS). Immune infiltration analysis revealed that the low-risk score group had a higher abundance of various immune cells, particularly cytotoxic T cells. Single-cell sequencing analysis of 17 NB tissues showed distinct immune cell distribution patterns in relation to TLS gene expression, with significant differences in neuron-like tumor cell and immune cell proportions between newly diagnosed and refractory relapsed NB samples. The invasion and migration ability of both NB cell lines decreased significantly after interference with CD200.

TLSs are localized immune response regions formed by CD20+ B cells surrounded by CD3+ T cells, resembling lymphoid follicles in SLOs (*Schumacher & Thommen, 2022*). Our research indicates that other cells may play a role in secreting key factors that promote the formation of TLS. For instance, we have found that smooth muscle cells, endothelial cells, and fibroblasts in NB tissue samples may express and secrete chemokines CCL2 and CCL21; monocytes and NK cells may express and secrete CCL4; and neuroblasts may express CD200, all contributing to the formation of TLSs within NB tissue. Comparable to the meshwork created by follicular dendritic cells (FDC) within SLOs, dense stromal networks serve to secure the position of TLSs in chronically inflamed tissues (*Sato et al., 2023*). These

TLSs are equipped with a unique vascular system consisting of high endothelial venules (HEV) that express peripheral node addressin (PNAd), which plays a crucial role in the recruitment of lymphocytes (*Teillaud et al., 2024*; *Schumacher & Thommen, 2022*). Despite the evident anatomical parallels between TLSs and SLOs, the majority of TLSs are not encapsulated within most tissues, facilitating the direct infiltration of their cellular constituents into the adjacent tissue (*Schumacher & Thommen, 2022*). This unimpeded access may lead to the exposure of immune cells within TLSs to the macromolecules present in the surrounding inflammatory milieu (*Li et al., 2023*). Additionally, it is currently believed that the molecular mechanisms and inducers of TLS and SLO formation are different, with the molecular inducers of TLSs being independent of the lymphotoxin signaling pathway (*Bar-Ephraim & Mebius, 2016*; *Neely & Flajnik, 2016*). The factors driving TLS generation are complex; for example, fibroblasts in the tissues of rheumatoid arthritis produce lymphochemokines such as CXCL13, CCL19, and CCL21, which are involved in TLS formation (*Wen et al., 2023*). Chemokines secreted by adipocytes and vascular smooth muscle cells can induce the formation of TLS in the mesenteric fat tissue of Crohn's disease patients and in atherosclerotic thrombotic arteries (*Guedj et al., 2019*). In our study, we identified four chemokines, CCL2, CCL4, CCL21, and CXCR3, in neuroblastoma samples, which may originate from different cellular subsets. Interestingly, single-cell analysis revealed that the sources and expression levels of these chemokines differ between newly diagnosed NB and relapsed NB groups. For example, CCL2 is primarily derived from fibroblasts and NK cells in newly diagnosed NB, while in refractory relapsed NB tissue, it originates from tissue stem cells, smooth muscle cells, and endothelial cells. Regardless, the expression levels of chemokines in newly diagnosed NB tissue are higher than in refractory relapsed samples. This may be related to the unique TME of the tumor, including pH, hypoxia, and vascular density. Chemokines produced by different cells may play different roles; some chemokines may collectively induce the recruitment of immune cells to the lymphoid domain and the vascularization of HEV, while others may favor the compartmentalization of lymphoid follicles (*Fleig et al., 2022*). Studies have found that different cytokines and chemokines can induce TLS with distinct characteristics (*Cabrita et al., 2020*; *Yin et al., 2023*). For example, tissue-specific expression of CXCL13 can induce the formation of TLS that aggregates B cells but lacks FDC interaction, while CCL21-induced TLS structures are larger and more organized (*Cabrita et al., 2020*; *Yin et al., 2023*; *Rouanne, Arpaia & Marabelle, 2021*). However, the specific roles of these prognostic chemokines in the formation and structure of TLSs in neuroblastoma remain understudied.

Notably, analysis of two independent multicenter datasets revealed that elevated CD200 mRNA levels correlated with improved NB prognosis. One interesting thing is that we found in two different medical center source datasets that the higher the mRNA level of CD200, the better the prognosis for NB. The CD200 expression has been detected in a variety of immune cells and normal tissues, including human thymocytes, neurons, activated T cells, B cells, dendritic cells, *Liu et al. (2020)*. CD200 is now more commonly known as an immune checkpoint molecule for immunosuppressive function, that works mainly by combining with CD200R (*Liu et al., 2020*; *Tang et al., 2024*). Our experiment found that the invasion and migration ability of NB cells down-regulated by CD200 was

significantly reduced. This suggests that CD200 may play a completely different role outside of NB cells than in the tumor cells themselves. Our experiments are in part consistent with the majority of research finding that CD200 expression on human cancer cells is thought to have a pro-tumor effect in cancer development (*Khan et al., 2021*). One study found that CD200/CD200R signal transduction promotes skin squamous cell carcinoma invasion and metastasis through ctsk expression (*Khan et al., 2021*). Nevertheless, the current study also suggests anti-tumor functions associated with CD200/CD200R pathway. In a 4THM breast carcinoma murine model, CD200 overexpression in CD200 transgenic BALB/c mice was correlated with the complete regression of primary tumors (*Erin et al., 2015*). CD200 has also been shown to have a protective effect in melanoma and non-small cell lung cancer (*Talebian et al., 2012*). Moreover, studies have shown that different CD200 genotypes have different prognostic effects on tumor patients. Such as rs1131199 GG genotype negatively influenced in the mortality of MM (*Gonzalez-Montes et al., 2024*). Therefore, some studies believe that CD200 may have dual effects on different links of different cancers (*Nip, Wang & Liu, 2023*).

TLSs have been found to be associated with good prognosis in many cancers, and their prognostic value is usually independent of the tumor's TNM staging (*Schumacher & Thommen, 2022*). Despite this, in breast cancer, a positive correlation has been found between the presence of TLSs and early tumor TNM stage (*Wang et al., 2022*; *Cabrita et al., 2020*). Our study found that in NB INSS staging, stage 3/4 NB samples have higher TLS-risk scores, and clinical high-risk group NB patient samples also have higher scores, indicating a poorer prognosis. Additionally, our subgroup analysis found that lower TLS scores correspond to higher mRNA levels of GZMB, PRF1, IFNG, and NK/T cytotoxic effector cytokines. A limitation of this study is the small number of clinical samples used for validation, which may lead to biased results. In our preliminary experiments, we were unable to identify typical TLS in NB tumor tissue samples through multicolor immunohistochemistry, which may be related to the characteristic reduced immune cell infiltration in NB. Furthermore, whether the key scientific hypothesis that exogenous supplementation of NB-TLS-related chemokines can promote the formation of TLSs within tumors and thus improve NB prognosis remains to be validated in our future efforts. Finally, and most importantly, since this study is an early exploratory study, including a wide range of NB patient samples rather than high-risk NB samples in clinical stages may limit the clinical reference value of this study. This requires us to conduct TLS research separately for the samples of clinical high-risk NB patients in the later stage.

## CONCLUSION

Our study demonstrates that tertiary lymphoid structure (TLS)-related genes play a crucial role in neuroblastoma (NB) prognosis, with a 6-gene TLS signature (CCL2, CCL4, CCL21, CD200, CXCR3, and IGSF6) serving as a promising prognostic biomarker for NB. CD200 may be a potential target for inhibiting the biological behavior of NB cells.

## ACKNOWLEDGEMENTS

Thank you, Dr. Ouyang from Jinan University, for the technical support related to bioinformatics analysis.

### Funding

This work was supported by the Fundamental Research Funds for the Central Universities (No. 2162431). The funders had no role in study design, data collection and analysis, decision to publish, or preparation of the manuscript.

### Grant Disclosures

The following grant information was disclosed by the authors:
Fundamental Research Funds for the Central Universities: 2162431.

### Competing Interests

The authors declare that they have no competing interests.

### Author Contributions

- Xuelian Liu performed the experiments, analyzed the data, authored or reviewed drafts of the article, and approved the final draft.
- Jian Deng performed the experiments, analyzed the data, authored or reviewed drafts of the article, and approved the final draft.
- Bingqing Yu performed the experiments, analyzed the data, authored or reviewed drafts of the article, and approved the final draft.
- Jiaxiong Tan conceived and designed the experiments, performed the experiments, analyzed the data, prepared figures and/or tables, authored or reviewed drafts of the article, and approved the final draft.
- Xiaoliang Lu conceived and designed the experiments, analyzed the data, authored or reviewed drafts of the article, and approved the final draft.
- Minmin Zhang conceived and designed the experiments, analyzed the data, prepared figures and/or tables, and approved the final draft.

### Data Availability

The raw data is available in the Supplemental File.

### Supplemental Information

Supplemental information for this article can be found online at http://dx.doi.org/10.7717/peerj.19767#supplemental-information.

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
