# Peer review of "Tertiary lymphoid structures-driven immune infiltration patterns and their association with survival in neuroblastoma"

_PeerJ, doi:10.7717/peerj.19767_

## Round 0.1 · original submission · Major Revisions

All reviewers raised serious concerns that require very careful analysis. Please address the critiques of the reviewers and amend the manuscript accordingly.

·

Basic reporting

Liu X et al report a mostly bioinformatics analysis of gene expression of TLS in neuroblastoma. Most of the study is taken from adult concepts and applied to neuroblastic tumors (NBTs) as if it was the same. To begin with, as the authors admit (at the end), there are no TLS in NBTs, therefore, the study is somewhat misfolded. Most importantly, NBTs are well known to include about 50% of low-risk and 50% of high-risk, which are relatively easy to discriminate with currently abundant markers. There is no need for further markers to distinguish low and high-risk NBs. What is most needed is how to distinguish among the high-risk NBs. Almost any biomarker study would come out positive to distinguish low and high-risk NBs. The results presented here are of low interest in this regard. Furthermore, there is little, if any, experimental confirmation of the in silico findings, which further more the interest of the article.

Experimental design

This is mostly a bioinformatics exercise of published databases. The analysis, however, includes a broad spectrum of NBTs, which diminishes the value of the results.

Comments:
1. Introduction. Authors claim the need for novel biomarkers to stratify NBTs. Is it really necessary? We have enough markers already to distinguish low and high-risk NBts. The real question should be: how to distinguish among high-risk NBTs? Furthermore, stage 3 cases are confusing and most of the time include a mixed bag of low and high-risk biological NBTs.

2. TLS components are not present in NBTs. Therefore, is the analysis of expression genes of Lymphoid structures is it adequate?

3. Introduction: Several language-based grammatical errors need correction.

Validity of the findings

The interest of the findings is questionable given the design of the analysis.

There is no formal and thorough validation of findings.

Additional comments

Conclusion section: statements are wrong and not supported by the data presented.
Language should be further reviewed.

Reviewer 2 ·

Basic reporting

The language used in the article is generally clear, with some exceptions in specific experimental sections, such as the real-time PCR and transwell migration assays. These sections do not adhere to the standard protocols typically employed in these experiments, as detailed below. The literature review is comprehensive, and the authors provide a well-defined background for their study.

Experimental design

Besides, there are several experimental inconsistencies that need to be addressed:
1. Concentration of CD200 siRNA: The authors should clarify the concentration of CD200 siRNA used in their experiment (line 215). Additionally, they should provide details on the number of cells used in the transwell migration experiments (line 221), including the surface area or plate type used.
2. Choice of Downregulated Gene: It is unclear why the authors chose to downregulate only CD200 using siRNA while not addressing other genes they identified, such as CCL2. This choice should be explained in the manuscript.
3. RNA Extraction Method: The RNA extraction method (lines 190-213) needs to be revised. While the authors mention using a total RNA purification kit, they do not specify the manufacturer, making it difficult for readers to assess the quality and reliability of the method. Furthermore, the description of the RNA extraction process (upper and lower liquid phases) is inconsistent with standard protocols and requires clarification.
4. RNA Concentration: The RNA concentration of 70 ng/mL (line 202) seems too low for reverse transcription, especially with the 30 µL reaction volume. Given this low concentration, the total RNA input for the reaction would be insufficient for effective reverse transcription. The authors should review these figures and provide more accurate details.
5. In line 209, the word “subsequently” should be removed, as reverse transcription is already mentioned in line 207.
6. For the figures, it is recommended that the authors ensure consistency in the Y-axis range across all bar graphs to improve the readability and clarity of the statistics. For example, the Y-axis range in Figure 7 G-L should be consistent, with a clear limit, such as 15, to ensure uniformity.

Validity of the findings

The authors used machine learning to identify six gene markers related to tertiary lymphoid structures (TLS) that could potentially serve as biomarkers for improved survival in neuroblastoma (NB) patients. While this work is promising and explores the application of machine learning to identify these markers, there are several significant concerns that must be addressed before the manuscript can be considered for publication.
Experimental Validation of Markers:
In the abstract, the authors mention establishing TLS as a new biomarker. However, the manuscript does not provide sufficient evidence to support how these six markers could be translated into reliable biomarkers. For example, although Kaplan-Meier plots show improved survival for patients with these markers, the manuscript does not explain how these results translate into clinical benefits, such as a reduction in disease progression. To substantiate their claim, the authors should provide additional data correlating these markers with clinical outcomes or risk stratification in neuroblastoma patients. Without this, it is difficult to accept these markers as robust biomarkers.
For instance, in the CD200 downregulation experiment (Figure 7), the authors show that CD200 downregulation decreases neuroblastoma cell invasion and migration. This finding contradicts the identification of CD200 as part of the six-gene set associated with improved survival, as it implies that CD200 may play a role in metastasis, which is inconsistent with their claims of improved survival.
Additionally, while the authors mention the lack of evidence regarding the presence of TLS in neuroblastoma tissues (lines 68-70) and its relationship with survival prognosis, they do not provide direct evidence for the presence of TLS in neuroblastoma. A more thorough focus on establishing the presence of TLS using their own data would strengthen the manuscript and help validate their claim that TLS is present and relevant in neuroblastoma prognosis.

Annotated reviews are not available for download in order to protect the identity of reviewers who chose to remain anonymous.

Reviewer 3 ·

Basic reporting

-

Experimental design

-

Validity of the findings

-

Additional comments

This manuscript investigates prognostic factors in neuroblastoma, focusing on tertiary lymphoid structures (TLS) using a large dataset. While the study is somewhat interesting, there are several concerns that need to be addressed. As the analysis is based on a dataset, careful attention must be paid to potential selection bias. In neuroblastoma, there are substantial biological differences among low-, intermediate-, and high-risk groups. Therefore, analyzing these groups collectively may have limited significance. Moreover, since it is already well established that patients with low- and intermediate-risk neuroblastoma generally have excellent prognoses, the clinical relevance of including these groups in the analysis is questionable. If the study cannot demonstrate that TLS-related factors stratify outcomes within the high-risk group, the overall clinical value of the findings would be limited. Thus, I strongly suggest performing an additional analysis focusing specifically on the high-risk group.

---

## Round 0.2 · accepted · Accept

All concerns of the reviewers were addressed and revised manuscript is acceptable now.

Reviewer 3 ·

Basic reporting

The authors responded to reviewer's comment appropriately.

Experimental design

The authors responded to reviewer's comment appropriately.

Validity of the findings

The authors responded to reviewer's comment appropriately.

Additional comments

The authors responded to reviewer's comment appropriately.